# Structure and Properties of Heat-Resistant Alloys NiAl–Cr–Co–*X* (*X =* La, Mo, Zr, Ta, Re) and Fabrication of Powders for Additive Manufacturing

**DOI:** 10.3390/ma14123144

**Published:** 2021-06-08

**Authors:** Vitaliy V. Sanin, Yury Yu. Kaplansky, Maksym I. Aheiev, Evgeny A. Levashov, Mikhail I. Petrzhik, Marina Ya. Bychkova, Andrey V. Samokhin, Andrey A. Fadeev, Vladimir N. Sanin

**Affiliations:** 1Scientific-Educational Center of SHS, National University of Science and Technology “MISiS”, Leninsky Prospect 4, 119049 Moscow, Russia; kaplanskii.ii@misis.ru (Y.Y.K.); levashov@shs.misis.ru (E.A.L.); petrzhik@shs.misis.ru (M.I.P.); bychkova@shs.misis.ru (M.Y.B.); fadeevandrei@gmail.com (A.A.F.); 2A.A. Baikov Institute of Metallurgy and Materials Science, Russian Academy of Sciences, Leninsky Prospect 49, 119334 Moscow, Russia; samokhin@imet.ac.ru; 3A.G. Merzhanov Institute of Sructural Macrokinetics and Materials Science, Russian Academy of Science, Acad. Osipyan Str. 8, Chernogolovka, 142432 Moscow, Russia; svn@ism.ac.ru

**Keywords:** heat-resistant alloys, NiAl, centrifugal SHS casting, hierarchical structure, plasma spheroidization

## Abstract

The NiAl–Cr–Co–*X* alloys were produced by centrifugal self-propagating high-temperature synthesis (SHS) casting. The effects of dopants *X* = La, Mo, Zr, Ta, and Re on combustion, as well as the phase composition, structure, and properties of the resulting cast alloys, have been studied. The greatest improvement in overall properties was achieved when the alloys were co-doped with 15% Mo and 1.5% Re. By forming a ductile matrix, molybdenum enhanced strength characteristics up to the values σ_ucs_ = 1604 ± 80 MPa, σ_ys_ = 1520 ± 80 MPa, and ε_pd_ = 0.79%, while annealing at T = 1250 ℃ and t = 180 min improved strength characteristics to the following level: σ_ucs_ = 1800 ± 80 MPa, σ_ys_ = 1670 ± 80 MPa, and ε_pd_ = 1.58%. Rhenium modified the structure of the alloy and further improved its properties. The mechanical properties of the NiAl, ZrNi_5_, Ni_0.92_Ta_0.08_, (Al,Ta)Ni_3_, and Al(Re,Ni)_3_ phases were determined by nanoindentation. The three-level hierarchical structure of the NiAl–Cr–Co+15%Mo alloy was identified. The optimal plasma treatment regime was identified, and narrow-fraction powders (fraction 8–27 µm) characterized by 95% degree of spheroidization and the content of nanosized fraction <5% were obtained.

## 1. Introduction

Nickel aluminide NiAl-based alloys are promising for designing the next-generation aircraft and aerospace products characterized by low specific weight and improved heat resistance, elevated-temperature strength, and creep resistance in the temperature range of 700–1100 °C [1,2,3,4]. The challenges related to application of the conventional casting technologies for product manufacturing because of low fracture toughness and difficulties with subsequent mechanical machining are the factors constraining the practical use of NiAl-based materials [1,4,5,6].

Recent advances in selective laser melting (SLM) or laser powder bed fusion (LPBF) makes it possible to fabricate complex-shaped items using NiAl-based intermetallic materials [7,8,9,10,11,12,13,14,15,16,17,18,19]. Narrow-fraction spherical powders of a heat-resistant alloy are used as a starting material in SLM. There are strict requirements to the chemical composition, impurity content and properties (bulk density, flowability, granulometric composition, and sphericity). Self-propagating high-temperature synthesis (SHS) and the technologies of elemental SHS [20], centrifugal SHS casting [21,22] in particular, are the efficient methods for producing ingots and powders from intermetallic alloys. Two integrated technologies have been developed earlier; they involve the following stages:(I)synthesizing the alloy from oxide raw materials by centrifugal SHS casting, vacuum induction melting (VIM) of billets (SHS ingots), casting of long size electrodes, plasma rotating electrode process (PREP), classification of narrow-fraction spherical powder, and SLM of complex-shaped products followed by hot isostatic pressing (HIP) [23,24,25,26,27];(II)elemental synthesis of the alloy, disintegration of the sinter cakes, classification and isolation of the target powder fraction, plasma spheroidization of the narrow-fraction powder, and near-net-shape part design by SLM followed by HIP [28,29,30,31].

Optimization of the composition and manufacturing regimes have made it possible to develop the β-NiAl-based alloy in the NiAl–Cr–Co–Hf system. The CompoNiAl–M5–3 alloy with composition NiAl–12%Cr–6%Co–0.25 at %Hf and a hierarchical structure is characterized by high strength in the temperature range between room temperature and 800 °C. The first structural level is formed by NiAl grains sized less than 40 µm with incoherent micron-sized α-Cr precipitates residing along the grain boundaries. The second level is formed by strengthening submicron-sized α-Cr and Hf precipitates inside the NiAl grains. The third level is formed by subgrains with coherent α-Cr precipitates smaller than 45 nm and the Ni_2_AlHf Heusler phase (3–5 nm in size) [24,25].

The use of spheroidized powders made of this alloy has allowed one to fabricate defect-free near net-shaped products (a turbine rotor blade) by SLM coupled with HIP [31,32]. The effect of post-treatment regimes on the structure and thermomechanical behavior was studied. Comparative uniaxial compression tests of the SLM samples after aging and HIP were conducted at a strain rate of 0.001 s^−1^ in the temperature range of 873–1373 K. Gasostatic treatment of the as-SLM parts made of the CompoNiAl-M5-3 alloy reduced porosity to 0.25 ± 0.08%, increased grain size and resistance to viscoplastic strain due to inhibition of Coble creep. The CompoNiAl-M5-3 alloy exhibits the following physico-mechanical properties that depend on the manufacturing method, technological conversion stage, and test temperature:


*The as-SHS ingot:*


at *T*_room_, *σ*_ucs_ = 2280 ± 260 MPa, *σ*_ys_ = 1670 ± 80 MPa, *E* = 175 GPa, *ε*_pd_ = 5%

800 °C: *σ*_ucs_ = 398 MPa, *σ*_ys_ = 345 MPa, *E* = 80 GPa, *ε*_pd_ = 45%


*after VIM of the as-SHS ingots:*


*T*_room_: *σ*_ucs_ = 1720 ± 90 MPa, *σ*_ys_ = 1150 ± 130 MPa, *E* = 160 GPa, *ε*_pd_ = 2%

800 °C: *σ*_ucs_ = 480 MPa, *σ*_ys_ = 420 MPa, *E* = 82 GPa, *ε*_pd_ = 48%;


*HIP of the powder atomized by PREP:*


*T*_room_: *σ*_ucs_ = 2440 ± 96 MPa, *σ*_ys_ = 1253 ± 25 MPa, *E* = 180 ± 3 GPa, *ε*_pd_ = 15 ± 1%

800 °C: *σ*_ucs_ = 503 MPa, *σ*_ys_ = 287 MPa, *E* = 121 GPa, *ε*_pd_ = 58%


*HIP of the spheroidized SHS powder:*


*T*_room_: *σ*_ucs_ = 2870 ± 67 MPa, *σ*_ys_ = 1131 ± 13 MPa, *E* = 181 ± 2 GPa, *ε*_pd_ = 17 ± 1%

800 °C: *σ*_ucs_ = 587 MPa, *σ*_ys_ = 395 MPa, *E* = 135 GPa, *ε*_pd_ = 63%;


*SLM + HIP + heat treatment (HT)*


*T*_room_: *σ*_ucs_ = 3050 ± 105 MPa, *σ*_ys_ = 1180 ± 43 MPa, *E* = 181 ± 4 GPa, *ε*_pd_ = 12 ± 1%

800 °C: *σ*_ucs_ = 713 MPa, *σ*_ys_ = 495 MPa, *E* = 75 GPa, *ε*_pd_ = 57%,

where *σ*_ucs_ is the ultimate compression strength; *σ*_0.2_ is the offset yield stress; *E* is the Young’s modulus; and *ε*_pd_ is the degree of plastic deformation.

The *E* and *σ*_ys_ values at 800 °C for the as-SLM samples subjected to heat treatment (aging) were 58 GPa and 334 MPa, respectively, while for the SLM + HIP samples these values were 75 GPa and 495 MPa, respectively. High-temperature creep tests revealed that the working temperature increased by 80–85 °C and the creep activation energy was elevated due to HIP treatment of the as-SLM products. Therefore, HIP (and low-temperature aging in some cases) are important operations for the SLM products made of NiAl-based alloys [26,27,31,32].

The drawbacks of the integrated technology (1) for NiAl-based intermetallic alloys include the high cost of VIM and casting long electrodes, as well as the challenges related to performing the PREP process at high rotation speeds because of the brittle nature of the alloys. Technology (2) involving plasma spheroidization of powders is more promising. The initial powder can be produced using various methods, including centrifugal SHS casting. Intermetallic alloy ingots are brittle; they can be crushed, ground, and classified.

Although the developed NiAl–12%Cr–6%Co–0.25at.%Hf alloy is characterized by a relatively high level of high-temperature strength and creep resistance, it would be interesting to study the effect of La, Mo, Zr, Ta, and Re dopants on the microstructure and properties of the intermetallic alloy. The choice of modifying additives (*X*) for the NiAl–Cr–Co–*X* alloy was based on the results of earlier studies [3,33,34,35,36,37,38,39,40], where dopants were found to have a favorable effect on the structure and properties of intermetallic compounds. Doping has made it possible to modify the structure near the grain boundaries and bind the doping elements to form additional compounds. When added even at low quantities (0.2–0.3%), lanthanum simultaneously increased the strength and ductility of the alloy [41]. Molybdenum contributed to enhancement of strength and thermal stability of intermetallic alloys [38,42]. Creep resistance and resistance to sulfide corrosion increase at molybdenum content lower than 3% [35,42]. Zirconium contributed to size reduction of the structural components of the alloy, which increased its elevated-temperature strength; heat resistance was also enhanced due to the formation of a protective ZrO_2_ film on the surface [43,44]. Ductility increases, while the number of fatigue cracks decreases at Zr content <0.9 wt.%. However, when the alloy contained an elevated amount of oxygen impurity, it became more brittle with increasing Zr concentration as the ZrO_2_ oxide phase was precipitated along the NiAl grains.

Along with lanthanum and zirconium, tantalum is one of the key doping elements for intermetallic alloys. Doping with Ta (several %) increased creep resistance and fatigue crack growth resistance [45]. Tantalum exhibits a much stronger strengthening effect compared to zirconium but leads to embrittlement and ductility reduction at contents above 4 wt.%.

Rhenium has a favorable effect on mechanical properties of heat-resistant intermetallic alloys. However, because it is one of the rarest elements in the Earth’s crust and is very expensive, ≤1.5 wt.% of rhenium is added in order to increase resistance to high-temperature creep [43,44,45,46,47]. Therefore, La, Mo, Zr, Ta, and Re elements can have a favorable effect on physicomechanical properties of the NiAl-based alloy [1,2,3].

This study aimed at producing heat-resistant NiAl–Cr–Co–*X* alloys based on β-NiAl with dopants *X* = La, Mo, Zr, Ta, and Re by centrifugal SHS casting; analyzing their structure and properties; and experimentally choosing the most promising alloy for the technology of plasma spheroidization of powders.

## 2. Materials and Methods

The cast alloy was produced by centrifugal SHS casting [21]. The synthesis was performed on a radial centrifugal setup upon exposure to high gravity up to 300 g. The overall schematic diagram of the centrifugal setup used in this study was earlier presented in references [21,23]. Due to the design of the setup, it was possible to set the rotational speed of the centrifuge rotor in a controlled manner to ensure the target acceleration level. A distinctive feature of this technology is that the relatively cheap oxide raw material is used and high flame temperature (2100–3500 °C) is attained. The chemical flowchart for the basic composition of the alloy can be written as follows:NiO + Cr_2_O_3_ + Co_3_O_4_+ Al+ (DA)+ (FA) –> [NiAl-Cr-Co-(*X*)] + Al_2_O_3_,
where: DA (doping agent) is La, Mo, Zr, Ta, and Re; FA (functional additive) is CaF_2_, Na_3_[AlF_6_], etc.

Table 1 lists the grades and characteristics of the initial components for preparing exothermic powder mixtures. The dopants were added into the reaction mixture so that the basic alloy was obtained. Experiments on co-doping the alloy with molybdenum and rhenium were additionally performed.

The preparation scheme of exothermic mixtures involved drying the components in SNOL-type drying cabinets at 90 °C for 1 h, dosing the reagents, mixing, and placing the mixture into graphite molds. Mixing was performed in an MP4/0.5 planetary ball mill for 15–20 min; the drum volume was 1 L; the ball-to-powder weight ratio was 1:5. The combustion temperature of the mixtures was higher than the melting point of the final synthesis products, thus making phase segregation possible due to gravity separation of the molten metal and the cinder. The highest degree of conversion of the target product into a metal ingot was achieved at the optimal acceleration value. The calculated composition of the NiAl-Cr-Co-(*X*) alloys and concentrations of the doping agents (*X*) are listed in Table 2. Components Zr, Ta, Re, and La were added to the reaction mixture as pure elements, while molybdenum was added in the oxide form. Tantalum and rhenium were added into the mixture in the form of powders, while lanthanum and zirconium, as metal chips 1–2 mm long and ≤100 µm thick. Boron (0.01 wt.%) was added to all the alloys under study in order to improve their casting characteristics and increase ductility.

The effect of heat treatment on the microstructure and mechanical properties of the cast samples was studied by annealing the samples in an SShVL-0.6.2/16I2 vacuum pit-type furnace with a heat shield without a ceramic liner at temperatures T_1_ = 850 °C, T_2_ = 1150 °C, and T_3_ = 1250 °C and residual pressure of 0.066–0.106 Pa for 3 h. Heat capacity was calculated according to the increase in temperature of water in the calorimeter filled with water where the samples heated to 100 °C had been immersed. The temperature at which alloy melting started and ended was determined by the method of torsional vibrations on a system for melt viscosity measurements (a viscometer) using cylindrical samples 16 mm in diameter and 30 mm high [48].

Impurity analysis was performed on a Foundry-Master LABFoundry-Master OE750 optical emission spectrometer (Hitachi, Japan). Gas impurity contents were determined on a TC-436 analyzer, and carbon impurities were detected on a CS-230 IH (LECO) analyzer. The compression tests were conducted on a LF-100KN universal testing machine (Walter + Bai AG, Switzerland) according to the GOST standard 25.503-97. The disintegration of cast ingots involved stepwise milling in a VEB LKS5 jaw crusher and comminution in an Activator-4M planetary ball mill (Russia) to a particle size of ≤45 µm. The fine-grained fraction with particle size of 20–40 µm was separated by air classification on a Golf-2 laboratory-scale centrifugal classifier (GeFest, Moscow, Russia). The precursor powder was treated with a flow of thermal plasma generated by electric arc discharge. The surface of the spherical powder was cleaned to remove condensed nanoparticles by ultrasonic treatment of the powder in a liquid. The granulometric composition of the particles was determined by laser beam diffraction on an ANALYSETTE 22 MicroTec plus laser diffractometer (Fritch GmbH, Idar-Oberstein, Germany). Bulk weight and flowability were determined according to the GOST standards 16440-94 and 20899-94, respectively.

The phase composition was determined by X-ray diffraction analysis on a D2 PHASER diffractometer (Bruker AXS GmbH, Karlsruhe, Germany) using Cu-Kα radiation within the range of 2*θ* = 10–140°. The microstructural studies were conducted on an S-3400N scanning electron microscope (Hitachi, Tokyo, Japan) coupled with a NORAN System 7 X-ray Microanalysis System energy-dispersive spectrometer (Thermo Scientific, Waltham, Massachusetts, USA). The effect of annealing on the crystal structure and phase composition of the NiAl–Cr–Co+15%Mo alloy was studied by high-resolution transmission electron microscopy (HRTEM) on a JEM-2100 microscope (Jeol, Akishima, Japan) using a Gatan 650 Single Tilt Rotation Analytical Specimen Holder (Gatan Inc., Pleasanton, CA, USA). The elemental composition of the phases was measured by energy-dispersive X-ray spectroscopy (EDXS) in the scanning TEM (STEM) mode. Ultra-thin foils for HRTEM studies were prepared by ion etching on a PIPS II System setup (Gatan Inc., Pleasanton, CA, USA). The Young’s modulus, hardness, and elastic recovery for individual structural components were determined on polished samples by nanoindentation on a Nanohardness Tester (CSM Instruments, Peuseux, Switzerland) at a load of 20 mN, loading rate of 40 mN/min, and exposure to the load for 5 s. The indentation curves were processed according to standard test method ASTM E2546−15.

## 3. Results and Discussion

Thermodynamic analysis of the adiabatic combustion temperature (T_ad_) using the THERMO software [34] showed that T_ad_ was 2300–2400 °C for all the compositions under study, being noticeably higher than the melting point of the products. A video of the combustion process and visual inspection of the samples demonstrated that the target product is the cast alloy (an ingot).

Earlier, V. Sanin et al. [21] showed that when synthesizing cast metallic materials by centrifugal SHS casting, exposure to external forces generated in centrifugal setups is among the key tools for affecting structure formation and composition of the resulting products. Visual inspection of the samples synthesized at different acceleration values demonstrated that in the absence of acceleration (a = 1 g), the synthesis products are characterized by high porosity and contain a significant amount of oxide inclusions Al_2_O_3_ and gas pores 0.2–1.5 mm in size (Figure 1). Increasing acceleration generated by a rotating rotor of the centrifugal SHS setup [21] increases the degree of phase separation and reduces the bulk concentration of inclusions. However, at accelerations below a = 50 g, inclusions are retained within the bulk of the sample, while the upper portion of the ingot contains a large shrinkage cavity. The ingots fabricated in the acceleration range a = 150–300 g had no noticeable inclusions or residual porosity. The samples removed from the molds could be easily separated into two layers: the lower layer was the target alloy; the upper layer was corundum Al_2_O_3_.

A comparative data analysis revealed no significant differences in the degree of phase separation of the synthesis products and dispersion of the mixture during synthesis of the samples with basic composition [24] and the compositions analyzed in this study. However, slight variation in combustion velocity (*U_f_*) of the mixtures was detected.

Figure 2 shows the combustion velocity *U_f_* of the analyzed mixture compositions as a function of acceleration. The reported data are the average of three experimental values.

At least three ingots 80 mm in diameter and 25–30 mm high were synthesized for each composition, and samples for the subsequent tests were cut from these ingots. Curve analysis (Figure 2) and optical spectroscopy studies of the ingots fabricated at different accelerations inferred that the optimal acceleration is a = 150 ± 5 g. An analysis of optical images on the transverse and longitudinal cross-sections of the samples revealed no residual non-metallic Al_2_O_3_ inclusions. Small-sized Al_2_O_3_ inclusions resulting from incomplete phase separation were observed at lower *a* values. No noticeable structural changes occurred when the acceleration was increased above 150 ± 5 g.

Table 3 shows the chemical composition of the target products of synthesis of multi-component NiAl–Cr–Co–*X* alloys; Table 4 lists the impurity contents. An analysis of the data demonstrated that concentrations of the main components and dopants are close to the calculated ones (Table 1). The only exception is lanthanum whose content in the target product was significantly lower than the calculated value. The noticeable deviation from the calculated value can be attributed to the fact that due to the high affinity for oxygen, most of La participated in the reaction of oxide reduction and competed with the main reducing agent (Al). The Al_2_O_3_-based cinder phase contained 0.3 wt.% of La. Therefore, doping with lanthanum exhibited little effect for fabricating the alloy by centrifugal SHS casting.

During any metallurgical process, ingots always contain impurities. SHS metallurgy is not an exception: the target product contains impurities even when the synthesis regimes have been properly chosen, so their mechanical properties can be deteriorated [1,21,39,40]. Thus, the ingots contain up to 0.1% of iron impurity due to the presence of 0.5 wt.% Fe in the initial powder of nickel oxide NiO.

Figure 3 and Figure 4 and Table 5 show the X-ray diffraction (XRD) data of the synthesis products and the mechanical properties of individual phases determined by selective indentation. β-NiAl is the main phase for all the synthesized compositions. The Ta-doped cast alloy contains the intermetallic compound Ni_3_(AlTa), while doping it with Zr yields the ZrNi_5_ phase.

An analysis of the indentation curve and the XRD data obtained by studying the NiAl–Cr–Co alloys doped with Mo, Zr, Ta, and Re revealed the typical mechanical properties of the NiAl, ZrNi_5,_ Ni_0.92_Ta_0.08_, and (Al,Ta)Ni_3_ phases, as well as the hypothetical Al(Re,Ni)_3_ phase, which was not detected by XRD. The values of these properties are summarized in Table 5.

When performing an analysis of the indentation data, we selected similar curves and classified them into groups in accordance with the experimentally determined phase composition of the alloy. The averaged hardness and Young’s modulus were then calculated for each phase. When doing so, it was taken into account that the intermetallic compound NiAl was the predominant phase (content >85%). Figure 4 shows an example of this approach. One can see that in addition to the curves typical of the NiAl phase (with the indent depth of ~330 nm), the doped NiAl–Cr–Co+0.5%Zr sample has another two indentation curves characterized by indent depth of >430 nm. This more ductile phase can presumably correspond to Ni-rich ZrNi_5_ intermetallic compound.

Figure 5, Figure 6, Figure 7, Figure 8 and Figure 9 show the microstructures and distribution maps of doping agents. An analysis of the microstructure and maps of element distribution in the alloy containing 2.5 wt.% Mo (Figure 5) revealed that interlayers of (Cr_0.8_Mo_0.2_) solid solution are formed between the branches of NiAl dendrites. A more detailed analysis of the microstructure showed that Mo is contained in the solid solution of chromium (6–16 at. %). Furthermore, molybdenum forms ductile precipitates with composition Cr_0.5_(Mo,Ni,Co)_0.5_ up to 20 µm long and up to 5 µm thick (light-colored areas). The (Cr_0.8_Mo_0.2_) interdendritic interlayers were found to contain submicron-sized NiAl-based precipitates (dark-colored inclusions), which resulted from concentration stratification of the mixed-composition phase under study with respect to nickel and aluminum.

Since microcracks propagate in intermetallic materials predominantly at intergrain boundaries via a longer path (to go round inclusions) [15,16], it is fair to assume that the alloy containing molybdenum precipitates will be characterized by increased ductility at room temperature.

The intergrain space of the Zr-doped alloy (Figure 6) was found to contain zone segregations of Ni5Zr. The characteristic size of the structural components and the principle of microstructure formation are similar to those for the developed Hf-doped CompoNiAl-M5-3 alloy [23,24,25,26]. It can be assumed that this microheterogeneous structure will not cause deterioration of mechanical properties.

An analysis of the microstructures of the Ta-doped alloy (Figure 7) showed that the intergrain space contains precipitates of the Laves phase (Cr_2_Ta). The Ni_3_AlTa phase is supposedly formed along the grain boundaries (the light-gray areas). Figure 8b shows the results of microprobe analysis along the line within the intergrain space. One can see that interlayers of Cr-based solid solution Cr(Co,Ni,Al) reside between NiAl grains.

The analysis of the microstructures of the Re-doped alloy (Figure 9) showed that size of all the structural components decreased twofold compared to that for other alloys. Doping heat-resistant alloys with rhenium especially affects their structural stability at high temperatures, behavior of the material upon primary creep, and oxidative characteristics [3]. Doping with rhenium to produce alloys using the centrifugal SHS casting technology was investigated for the first time in this study. Elemental analysis demonstrated that the added amount of the component completely passes into the metal ingot, without any losses for formation of the cinder phase. It is important to mention that rhenium is homogeneously distributed over the intergrain space of the Cr-based solid solution. The content of rhenium in the solid solution ranges from 2 at. % (dark gray regions) to 12 at. % (light-colored regions) (Figure 9).

Taking into account the features of structure formation of the molybdenum-containing phases, additional experiments were conducted, aiming to synthesize the NiAl–Cr–Co alloy with increased Mo concentration (up to 15%) and co-doping it with molybdenum and rhenium (15 wt.%Mo + 1.5 wt.%Re). In the former case, the objective was to increase the ductility of the alloy at room temperature by increasing the bulk content of the ductile (Cr, Mo) phase. Additional doping with rhenium was performed to obtain a fine-grained structure. Table 6 summarizes the elemental composition of the NiAl–Cr–Co+15%Mo and NiAl–Cr–Co+15%Mo+1.5%Re alloys, while Table 7 lists the impurity content.

The XRD pattern of the synthesis products are shown in Figure 10. β-NiAl, (Cr, Mo) solid solution, and the accompanying phases shown in Table 8 are the main phases for both compositions. The sample with 15 wt.%Mo + 1.5 wt.%Re contains the Laves phase MoRe_2_.

The results of microstructural studies of the alloys with increased Mo and Mo + Re contents are shown in Figure 11 and Figure 12, respectively.

The alloy doped with 15% Mo has a cellular structure (Figure 11) and consists of the following phases: NiAl, Cr-based solid solution, Mo-based solid solution, and the (Ni,Cr,Co)_3_Mo_3_C phase that was identified by a more thorough study.

Precipitates of the MoRe_2_ phase are additionally observed within the structure of the alloy doped with 15%Mo + 1.5%Re (Figure 12). It is expected that the dispersed Cr(Mo) and MoRe_2_ particles will have a favorable effect on strength properties of the alloy (and primarily on resistance to viscoplastic flow) due to deceleration of mobile matrix dislocations [38].

Table 9 summarizes the properties of SHS ingots: the melting point *T*_melt_, density *ρ*, heat capacity *C*_v_, hardness, ultimate compression strength *σ*_ucs_, the offset yield stress *σ*_ys_, and the degree of plastic deformation *ε*_pd_. Samples of the alloys co-doped with molybdenum and rhenium exhibited the highest strength characteristics (Figure 13a). Thus, the ultimate compressive strengths of the alloys doped with 2.5% Mo and 15% Mo were *σ*_ucs_ = 1586 and 1728 MPa, respectively, while *σ*_ucs_ of the alloy co-doped with Mo and Re was 1800 MPa. These values are comparable with strength of the CompoNiAl-M5-3 alloy subjected to vacuum induction melting [25]. The alloys doped with Re and Ta were also characterized by a relatively high ultimate compressive strength.

The resulting experimental data demonstrate that formation of the ductile phase in the intergrain space had a positive effect on mechanical properties, while the characteristics of the alloys can be further improved during the subsequent technological stages, including HIP [36] and SLM, in a manner similar to that for the CompoNiAl-M5-3 alloy.

Melt crystallization rates achieved during alloy production by centrifugal SHS casting can be as high as 20–25 °C/s [21]. Supersaturated solid solutions are formed at these crystallization rates of multicomponent melts; the phases exist in a non-equilibrium state; and the samples contain residual stresses. Therefore, vacuum annealing usually favorably affects strength and ductility [49]. Heat treatment of the NiAl–Cr–Co–Hf alloy at *T* = 850 °C and *p* = 10^−2^ Pa for 3 h simultaneously increased strength and ductility due to concentration stratification of the supersaturated chromium-based solid solution and precipitation of strengthening α-Cr nanoparticles (sized less than 45 nm) and the Heusler phase Ni_2_AlHf (3–5 nm) [24].

In this study, annealing was performed for the alloys doped with 15% Mo and 15%Mo + 1.5%Re and was shown to have a favorable effect on mechanical properties of the alloys. Figure 13b,c shows that strength and ductility increase noticeably as the annealing temperature rises from 850 to 1250 °C. There is a plastic deformation region at 1250 °C, corresponding to residual deformation *ε_pd_* = 2.01 and 6.15% for the alloys doped with 15%Mo and 15%Mo + 1.5%Re, respectively. It is important to mention that Re also increases the degree of plastic deformation because the grain structure of the alloy is refined as it is uniformly distributed along the intergrain boundaries predominantly within chromium- and molybdenum-based solid solutions.

Figure 13 compares our findings with the data reported in ref. [24] for the Ni_41_Al_41_Cr_12_Co_6_ alloy. The diagram was re-plotted with allowance for rigidity of the testing machine and the Young’s modulus determined graphically, which is equal to ~200 GPa for the NiAl-based alloys [1,4]. This alloy is characterized by high ultimate compressive strength *σ*_ucs_ = 2250 MPa. Annealing at 1250 °C made it possible for the alloy doped with 15% Mo to approach this value, while the alloy doped with 15%Mo + 1.5%Re has reached it (Table 10). Plastic deformation of the NiAl-Cr-Co+15%Mo+1.5%Re alloy annealed at 1250 °C is higher than that of the Ni_41_Al_41_Cr_12_Co_6_ alloy by 1.92% due to precipitation of the viscous (Cr,Mo) phase as interdendritic interlayers.

Unlike the integral mechanical properties evaluated during the compression tests, nanoindentation measurements showed that local mechanical properties (hardness H and the Young’s modulus E) as a function of annealing temperature were reduced by 10–12% (Figure 14). This can be possibly related to the coherence loss at the interface between the nanosized disc-shaped Cr-based precipitates and the supersaturated solid solution via the mechanism of Guinier–Preston structural transformation, which takes place in NiAl–Cr–Co–Hf alloys at temperatures above 850 °C as it was determined earlier in ref. [24]. Figure 14 shows that this thermal behavior of local properties is typical of the samples cut from the ingots both along and across their axis. This was proved by the texture formed during centrifugal SHS casting, which has led to anisotropy of the properties.

By comparing Figure 13 and Figure 14, one can infer that local disordering during annealing increases the content of the plastic component of strain ε_pd_ in the compression tests.

Figure 15 shows the microstructures of the NiAl–Cr–Co+15%Mo alloy before (a) and after annealing at 1150 °C (b) and 1250 °C (c).

One can see that the alloy structure is more homogeneous in the range of maximal working temperatures for this class of materials (1150 °C); NiAl grains become smaller. Growth of dendritic grains, the (Ni,Cr,Co)_3_Mo_3_C phase, and the (Cr,Mo) solid solution took place at annealing temperature of 1250 °C. The structural components of the NiAl–Cr–Co+15%Mo alloy after annealing at 1150 °C (a) and 1250 °C (b) are shown in Figure 16.

Submicron-sized NiAl precipitates are observed after annealing 1150 °C (Figure 16a). No dispersed Cr(Mo) particles were detected within the bulk of NiAl grains. Contrariwise, precipitates of the (Cr,Mo) phase distributed both within the dendrite bodies and along the boundaries of NiAl grains can be seen after annealing at 1250 °C (Figure 16b). None of the two samples contained the Mo-based solid solution phase after the annealing.

In order to investigate the crystal structure of the components of the NiAl–Cr–Co+15%Mo alloy before and after annealing at 1250 °C, we studied ultrathin foils made of this alloy by HRTEM and electron diffraction.

Figure 17a,b shows the bright-field (BF) TEM images of the characteristic structure of the NiAl–Cr–Co+15%Mo alloy near the interface boundary of a dendritic cell. According to the EDXS data, dendrites were composed of the solid solution of chromium and cobalt in *β*-NiAl (Table 11, spectrum 1). A feature of this material was that elongated grains of the molybdenum-containing phases (1–2 µm wide) (Table 11, spectra 2–5) were formed in the interdendritic space owing to the effect of non-equilibrium conditions of melt crystallization at excessive molybdenum and chromium contents during SHS casting [21,24]. The relatively slow cooling down of the ingots in air contributed to additional precipitation of nanosized (<100 nm) particles of the excessive phase (composition: Cr, 65.92 at.%; Mo, 25.39 at.%; Ni, 5.62 at.%; and Al, 2.07 at.%) in the dendrite bodies. These precipitates increased the resistance to plastic deformation due to dispersion strengthening of the NiAl matrix as previously demonstrated in references [31,36,50,51].

A more detailed analysis of the fine structure of dendritic cells revealed coherent (Cr,Mo) precipitates sized 10–20 nm, which is demonstrated by the HRTEM image of the *β*-phase taken along the [111] zone axis (Figure 17c). The unit cell parameter of NiAl calculated using the SAED pattern (the inset on the left-hand side in Figure 17c) was *a* = 2.952 Å, being higher than the tabulated value (*a* = 2.887 Å) by 2.3%. The increase in the unit cell parameter of the matrix phase was presumably caused by dissolution of molybdenum, chromium, and cobalt atoms in this phase as proved by the results of EDXS analysis of the chemical composition of the dendrite body (Table 11, spectrum 1). The crystal structure of dendritic cells was characterized by high number density of defects in the form of alternating dark- and light-contrast lines oriented along <200> (Ni, Fe)Al. These defects resulted from plastic deformation of the matrix by passing numerous dislocations of the crystal lattice on the <110> planes upon exposure of the alloy to compressive stress, which is proved by the presence of additional reflections in the SAED pattern (the inset in Figure 17c) exhibiting characteristic mirror-type dislocation with respect to the reflections with stronger intensity. The IFFT-filtered [001] HRTEM image of the crystal structure of the NiAl/(Cr, Mo) interface from the area indicated with A in Figure 17c demonstrates that there is an *a*/2{110} misfit dislocation (marked with T), thus indicating that even the excessive precipitates (≤14 nm in size) in the deformed alloy have lost complete coherence.

Figure 17d shows the HRTEM image of the crystal structure of the multicomponent phase in the Mo–Cr–Ni–Co system (Table 11, spectra 2 and 3). According to the quantitative ratio between the main components in the phase, we have put forward a hypothesis that a continuous series of solid solutions with a bcc structure has been formed. However, the detected Mo-based phase had an fcc crystal lattice oriented along the [114] zone axis. This was confirmed by the results of measuring angles between the main direction vectors in the SAED pattern (the inset in Figure 17d), which fully corresponded to the tabulated values and were equal to 63.9° and 50.1° [52,53]. The lattice parameter of the phase under study calculated using the recorded SAED pattern with allowance for the Miller indices was *a* = 11.37 Å. Taking into account the determined lattice parameters, one can state that a complex carbide (Ni,Co,Cr)_3_Mo_3_C crystallizes in the interdendritic space. Along with boron, the impurity carbon interacts with (Cr, Mo) at grain boundaries to give rise to carbides, thus being involved in dispersion hardening of the alloy by slowing down grain boundary diffusion [54,55]. The IFFT filtered image of the crystal structure (marked with B) shows the ordered arrangement of atoms of the second crystal lattice between the atoms of the first one, being indicative of superstructure formation via the defect-ordering mechanism [51,54].

Figure 17e shows the results of analyzing the crystal structure of a grain with composition Mo, 78 ± 2 at. % and Cr, 22 ± 2 at. % (Table 11, spectra 4 and 5) near the interface boundary with the dendritic cell of *β*-NiAl. According to the recorded SAED pattern (the inset in Figure 17e), we have identified the lattice type and parameters for the phase under study. Taking into account the angles between the main crystal directions (61.2° {2201} and 28.2° {2020}) of the hcp lattice and the calculated lattice constants (*a* = 5.92 Å and *c* = 6.41 Å), a complex boride (Mo_0.8_Cr_0.2_)_x_B_y_ was formed instead of (Mo,Cr) solid solution. Boron could not be identified at the stage of phase composition measurements, since an X-Max80 T silicon drift detector (Oxford Instruments, High Wycombe, UK) was used in the study.

The FFT image from *β*-NiAl dendritic cell (marked with C) compared to the [1216] SAED inset shows that there is no coherent bonding of crystal lattices of the phases at the interface with (Mo_0.8_Cr_0.2_)_x_B_y_ (Figure 17e), since the main crystal direction vectors do not coincide. Nevertheless, the analyzed phases have close orientation, so it might be possible to provide conditions when boride precipitates will be coherent. This will cause additional dispersion hardening of the material when elastic strain fields generated by the crystal lattices of two phases impede the gliding of matrix dislocations.

The IFFT filtered [1216] HRTEM image recorded for the area indicated with D in Figure 17e shows the high-density stacking faults oriented along the atomic planes {2201} (Figure 17f). This stacking fault (SF) with an additional similar layer was likely to be produced by aggregation of interstitial atoms [44]. Hence, plastic deformation of the (Mo_0.8_Cr_0.2_)_x_B_y_ phase exposed to compressive strain is accompanied by dissociation of total dislocations into partial ones.

The BF TEM image of the structure of the annealed NiAl–15Mo–12Cr–6Co alloy taken along the [111] (Ni,Co,Cr)_3_Mo_3_C zone axis is shown in Figure 18a. Annealing caused diffusion-controlled growth of strengthening Cr(Mo) nanoprecipitates along the intergrain boundaries up to the submicron size (150–400 nm) and increased chromium and cobalt concentrations in the (Ni,Co,Cr)_3_Mo_3_C phase (Table 11, spectrum 6). According to references [24,26,31,44], precipitation of dispersed particles of the excessive Cr(Mo) phase took place at the stage when the ingots were cooled down in the vacuum chamber of the electric furnace as a result of concentration stratification of supersaturated solid solution Cr(Mo) in the intermetallic matrix. Changes in the composition of interdendritic interlayers are caused by its diffusion saturation with doping elements as dispersed inclusions (Cr,Mo) are dissolved in *β*-NiAl during the heat treatment of the alloy. The [111] SAED pattern (marked with F in Figure 17a) proved that there is the (Ni,Co,Cr)_3_Mo_3_C phase (fcc, *a* = 10.74 Å) having a suprastructure. The decrease in the lattice parameter of the molybdenum-containing phase by 5.5% for the annealed alloy can be caused by the fact that molybdenum concentration in this phase is reduced.

No coherent coupling of atomic planes of crystal lattices between the main (Ni,Co,Cr)_3_Mo_3_C and *β*-NiAl phases has been detected as indicated by the [111] HRTEM image of (Ni,Co,Cr)_3_Mo_3_C crystal structure at the interface with the dendritic unit cell of the matrix phase (Figure 18b). Despite the deformed state of the alloy, no linear defects were revealed in the phase under study. The inset in Figure 18b also demonstrates the supra-structure of the crystal lattice of the phase being analyzed.

The bodies of the dendritic cells were found to contain highly dispersed (Cr,Mo) inclusions sized ~21 nm (Figure 18c). The observed coupling of atomic planes in the HRTEM image of (Cr,Mo)/NiAl interface taken along a common [111] zone axis, as well as the superposition of the diffraction spots in the FFT patterns for the matrix and the precipitates confirm that they are coherent. Therefore, the *β*-NiAl matrix phase and (Cr,Mo) precipitates exhibit the [111]*_β_* || [111] _Cr(Mo)_ and {110}*_β_* || {110}_Cr(Mo)_ orientation relationships between their crystal lattices. The development of plastic deformation in the alloy due to compressive stress resulted in accumulation of *a*/2[111] (011) edge dislocations formed by extra half-planes of the same sign as demonstrated by the IFFT filtered area marked with G in Figure 18c. This fact also explains the nature of the dark-colored nanosized zones in the crystal structure of the material, which mainly reside around a coherent precipitate of the excess phase; their presence is caused by elastic deformations of the lattice around the dislocation centers.

Further spheroidization studies were conducted for the most promising alloy with composition NiAl-Cr-Co + 15%Mo. The precursor powder having a target composition was obtained by mechanical grinding of the cast SHS ingots. The spherical powder for additive manufacturing SLM machines was fabricated using the plasma spheroidization method.

The precursor powder of the alloy under study had a characteristic fragmented morphology of particles (Figure 19a). Most particles were ≤30 µm in size; however, the powder also contained relatively coarse particles sized up to 50 µm. It is important to mention that the classified powder contained no submicron-sized particles that abruptly deteriorate the technological properties of the powder and have an unfavorable effect on the stability of plasma spheroidization, thus causing excessive evaporation and condensation of the material on the surface of coarse particles [30].

Figure 19b shows the integral and differential particle size distribution of the NiAl–Cr–Co+15%Mo powder after mechanical grinding in a planetary ball mill followed by air classification. Particle size of the precursor powder ranged between 7 and 79 µm. The average particle diameter D_av_ was 33.9 µm. The resulting powder was characterized by bimodal particle size distribution. The mode corresponding to the first peak was 17 µm, while the mode for the second (higher) peak was 49 µm. The distribution quantiles D_10_, D_50_, and D_90_ were 12.3, 31.6, and 60.7 µm, respectively. The content of the fraction <20 µm and >40 µm was 35% and 38%, respectively.

To choose the optimal powder spheroidization regime, we conducted a series of experiments aiming to evaluate the effect of enthalpy of plasma flow, the composition of the plasma supporting gas, and flowability of the precursor powder on the spheroidization degree and the intensity of powder evaporation yielding condensed nanoparticles. The study was conducted at the following design and technological parameters: plasma torch power *N*_pl_ = 6.6–13.8 kW; Ar and Ar + H_2_ were used as plasma supporting gases; the flowability of the plasma supporting gas *G*_pl.gas_ = 2.7 m^3^/h; the enthalpy of plasma flow *I*_pl_ = 1.04–1.91 kW h/m^3^; and powder feed rate *G*_powder_ = 1.5–6.0 kg/h.

Having analyzed the results of the studies, we found that the degree of spheroidization of the product increased from 80% to 92% with rising enthalpy of plasma flow. A powder with 65–99% degree of spheroidization can be produced depending on the feed rate of the precursor powder. The use of hydrogen-containing thermal plasma increases heat capacity in the “gas–processed material” system. This results in a higher rate of heating of the particles being processed and an increase in the degree of powder spheroidization to 99%. The intensity of powder evaporation depending on various treatment parameters was determined experimentally. The content of the nano-sized fraction in the powder ranged from 7.9 to 13.5 wt.%.

When using argon plasma with the powder feed rate of 3 kg/h and the enthalpy of plasma flow *I*_pl_ = 1.71 kW h/m^3^, the resulting powder was characterized by the degree of spheroidization of 92% and nanoparticle content of 10% (Figure 20a). Before conducting the structural analysis and measuring particle size distribution, the powder was subjected to ultrasonic treatment in a liquid to remove condensed particles from its surface.

Complete powder spheroidization was not achieved at these parameters of plasma treatment (irregular-shaped particles were present in the range of 10–20 µm and 30–50 µm). Most spherical particles contained Al_2_O_3_ inclusions caused by evaporation of the material, oxidation of aluminum, and condensation of the material on the surface. According to the laser diffraction data (Figure 20b), the spheroidized powder had a unimodal particle size distribution in the range of 5–44 µm. The characteristic dimensions were as follows: D_av_ = 18.0 µm; D_10_ = 8.4 µm; D_50_ = 16.4 µm; and D_90_ = 30.0 µm.

Due to the use of the Ar–H_2_ mixture as a plasma supporting gas, the enthalpy of plasma flow was increased to 1.9 kW h/m^3^ and the degree of powder spheroidization rose to 98% (Figure 21a) as energy density of the plasma flow per unit of surface area of the processed material was increasing. However, the content of condensed particles also rose to 11.6%. An analysis of the morphology of powder particles revealed no irregular-shaped sharp-cornered particles and a significantly smaller content of Al_2_O_3_ inclusions. Some particles contain satellites sized 1–10 µm. Figure 21b shows the granulometric composition of spherical particles at these process parameters. The increasing enthalpy of plasma flow made it possible to narrow the range of particle size distribution to 6–26 µm. The powder is characterized by unimodal particle size distribution with D_av_ = 13.53 µm. The distribution quantiles were as follows: D_10_ = 9.0 µm; D_50_ = 13.2 µm; and D_90_ = 18.7 µm.

The studies revealed the most efficient powder spheroidization regime. The 95% degree of spheroidization was achieved by using the optimal design and technological parameters; the content of the nano-sized fraction was 5%. The following process characteristics of the spherical powder were identified: flowability, 20.5 s; bulk density, 4.04 g/cm^3^. Figure 22c shows the granulometric composition of the resulting powder. The characteristics of the powder are as follows: D_av_ = 14.8 µm; D_10_ = 10.5 µm; D_50_ = 14.5 µm; and D_90_ = 19.7 µm. The powder is characterized by unimodal particle size distribution within the range of 8–27 µm.

Figure 22a,b shows the morphology of the resulting NiAl-Cr-Co+15%Mo alloy powder. Powder particles have a regular spherical shape, and almost no Al_2_O_3_ inclusions are detected.

The microstructure of the surface and the transverse cross-section of the NiAl-Cr-Co+15%Mo alloy powder after plasma treatment is shown in Figure 23. The powder has a characteristic dendritic grain structure with unit cell dimension of 0.2–3 µm; the grains are formed by supersaturated solid solution of dopants (Cr, Co, Mo) in the NiAl matrix. The interdendritic space contains continuous interlayers of Cr(Mo, Co) solid solution. The resulting powders will be used in further studies of the laser powder bed fusion (LPBF) process.

## 4. Conclusions

Alloys in the NiAl–Cr–Co–(*X)* system have been produced by centrifugal SHS casting. The effects of dopants *X* = La, Mo, Zr, Ta, and Re on combustion, as well as the phase composition, structure, and properties of the resulting cast alloys, have been studied. A eutectic cellular structure is formed in the alloys co-doped with Mo and Re. Co-doping with 15% Mo and 1.5% Re has ensured the greatest improvement in overall properties. In the alloy doped with 15% Mo, molybdenum forms a ductile matrix and enhances the strength characteristics up to the following values: *σ*_ucs_ = 1604 ± 80 MPa, *σ*_ys_ = 1520 ± 80 MPa, and *ε*_pd_ = 0.79%. Annealing at *T* = 1250 and *t* = 180 min improves strength characteristics to the following level: *σ*_ucs_ = 1800 ± 80 MPa, *σ*_ys_ = 1670 ± 80 MPa, and *ε*_pd_ = 1.58%. Rhenium modifies the structure of the NiAl–Cr–Co+15Mo1.5Re alloy and improves properties (*σ*_ucs_ = 1682 ± 60 MPa, *σ*_ys_ = 1538 ± 60 MPa, and *ε*_pd_ = 0.87%), while annealing additionally enhances them (*σ*_ucs_ = 2019 ± 60 MPa, *σ*_ys_ = 1622 ± 60 MPa, and *ε*_pd_ = 5.88%).The mechanical properties of the NiAl, ZrNi_5,_ Ni_0.92_Ta_0.08_, and (Al,Ta)Ni_3_ phases, as well as the hypothetical Al(Re,Ni)_3_ phase, have been determined by nanoindentation of the alloys. Local disordering upon annealing above 850 °C increases the rate of plastic deformation in the compression tests due to the coherence loss at the interface between nanosized disc-shaped Cr-based precipitates and supersaturated solid solution via the mechanism of Guinier–Preston structural transformation.The three-level hierarchical architecture of the NiAl–Cr–Co+15%Mo alloy has been identified: the first level is formed by dendritic *β*-NiAl grains with the interlayers of molybdenum-containing phases (Ni,Co,Cr)_3_Mo_3_C and (Mo_0.8_Cr_0.2_)_x_B_y_ (cell dimension < 50 µm); the second level is formed by strengthening submicron-sized Cr(Mo) particles distributed along the grain boundaries; and the third level consists of coherent Cr(Mo) nanoprecipitates (10–40 nm) within the bodies of *β*-NiAl dendrites.The optimal plasma treatment regime has been identified, and narrow-fraction powders (fraction 8–27 µm) characterized by 95% degree of spheroidization and the content of nano-sized fraction <5 have been obtained. The powder has a characteristic dendritic structure with the grain size of 0.2–3 µm.

## Figures and Tables

**Figure 1 materials-14-03144-f001:**
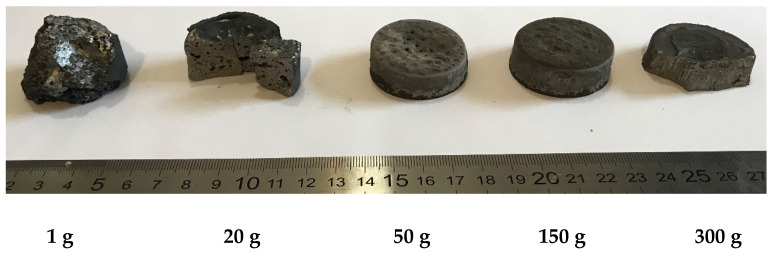
The appearance of NiAlCr(Co)-2.5%Mo alloy ingots produced at different acceleration values (g).

**Figure 2 materials-14-03144-f002:**
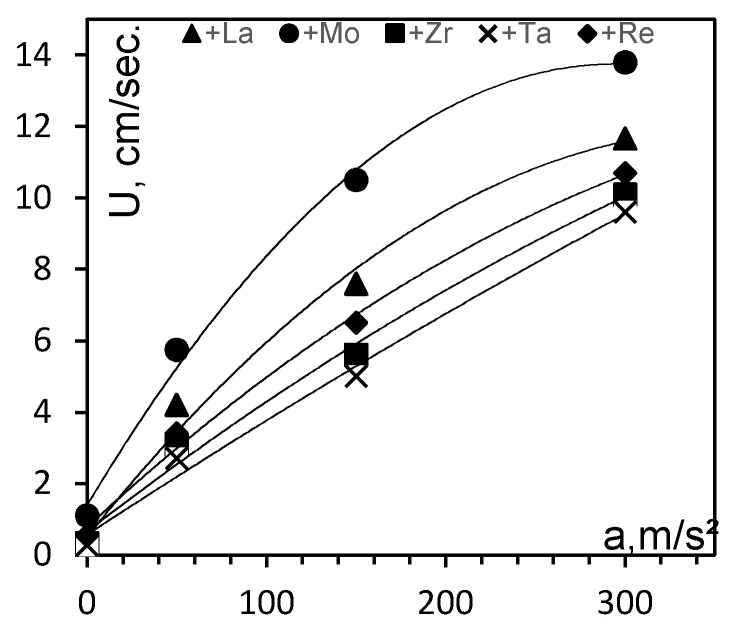
The effect of acceleration (*a*) and added modifying agents (*X*) on the combustion velocity (*U_c_*) of the mixtures during synthesis of NiAl–Cr–Co–*X* cast alloys, where *X* = La, 2.5%Mo, Zr, Ta, and Re.

**Figure 3 materials-14-03144-f003:**
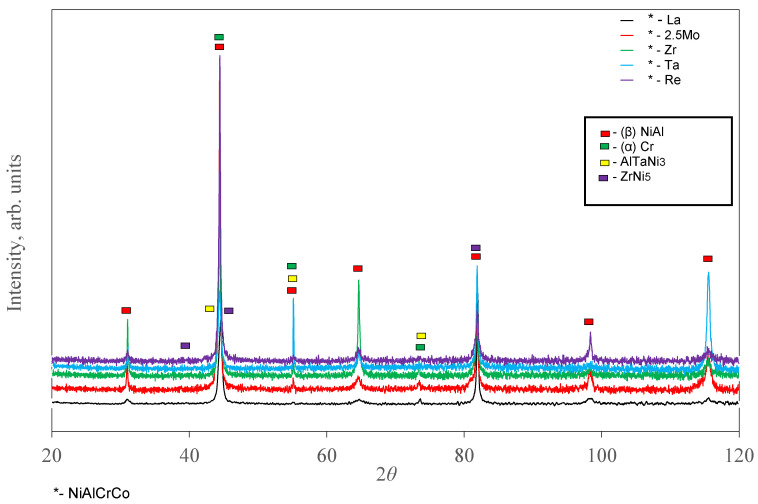
The X-ray diffraction (XRD) patterns of the NiAl–Cr–Co–(*X*) alloys, *X* = La, 2.5%Mo, Zr, Ta, and Re.

**Figure 4 materials-14-03144-f004:**
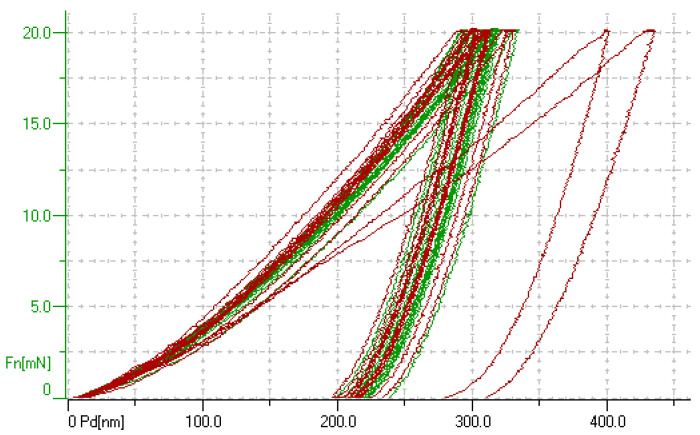
The experimental indentation curves of the single-phase alloy NiAl–Cr–Co+2.5%Mo and the double-phase alloy NiAl–Cr–Co+0.5%Zr.

**Figure 5 materials-14-03144-f005:**
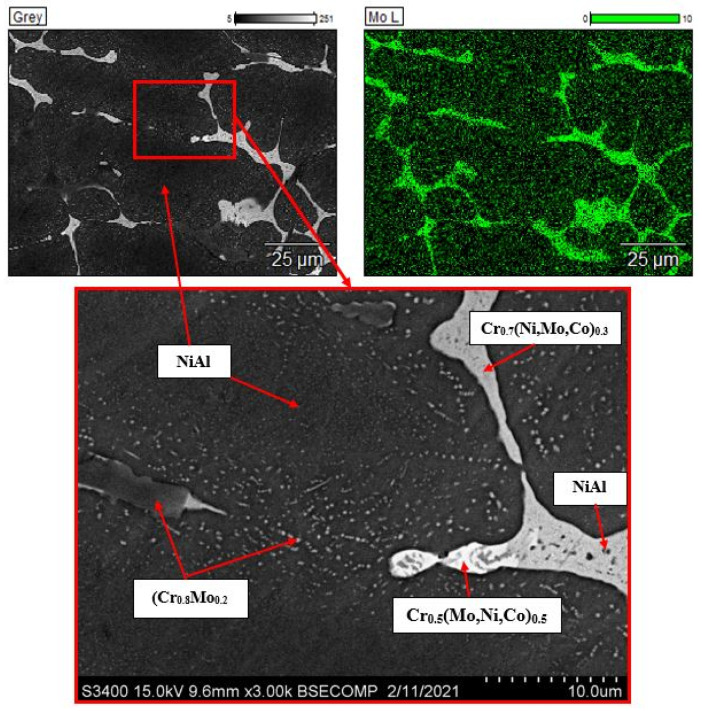
The microstructure of the alloy doped with 2.5% of molybdenum and the map of dopant distribution.

**Figure 6 materials-14-03144-f006:**
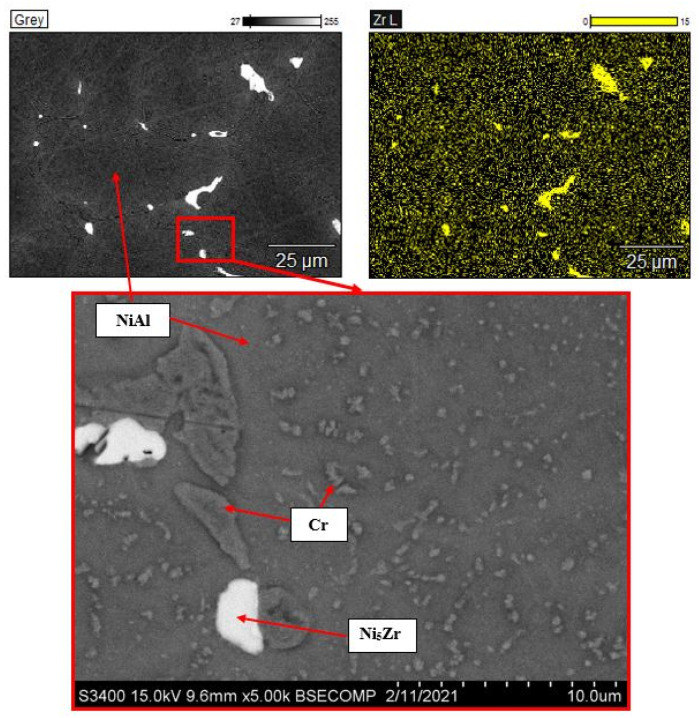
The microstructure of the alloy doped with 0.5% of zirconium and the map of dopant distribution.

**Figure 7 materials-14-03144-f007:**
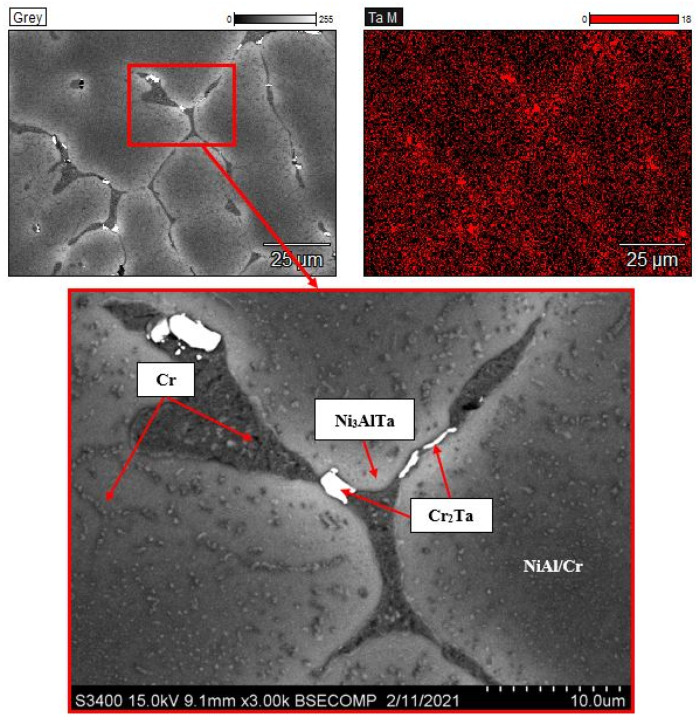
The microstructure of the alloy doped with 2.5% of tantalum and the map of dopant distribution.

**Figure 8 materials-14-03144-f008:**
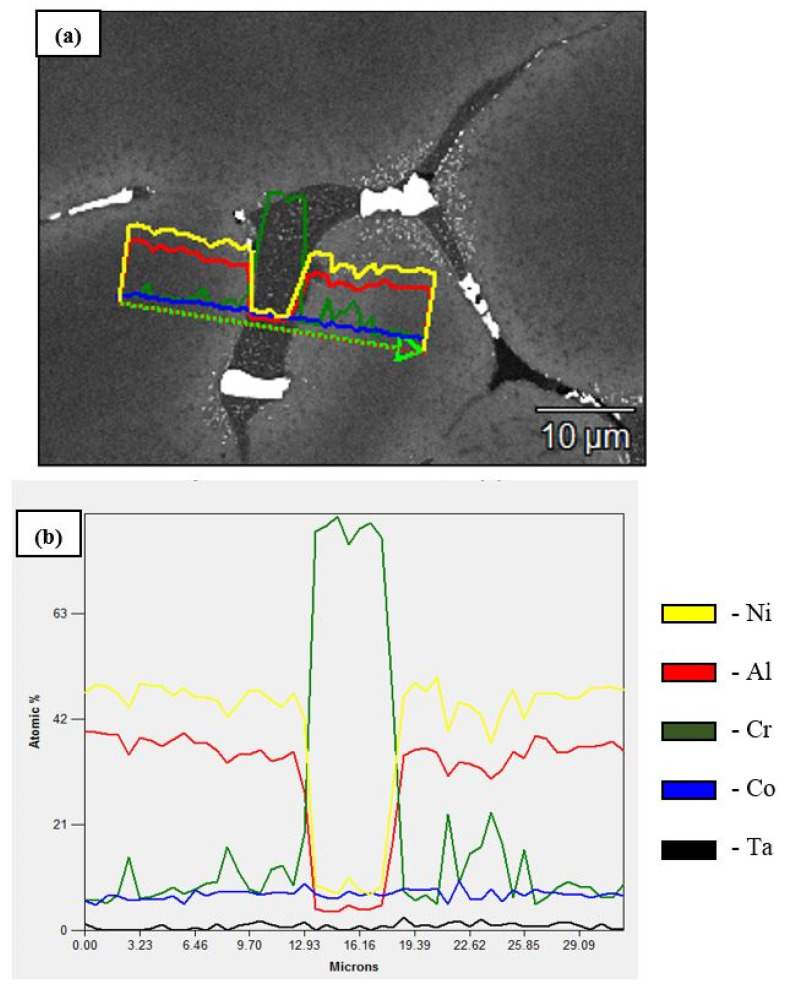
The microstructure of the alloy doped with tantalum (**a**) and the profile of element distribution in the intergrain space (**b**).

**Figure 9 materials-14-03144-f009:**
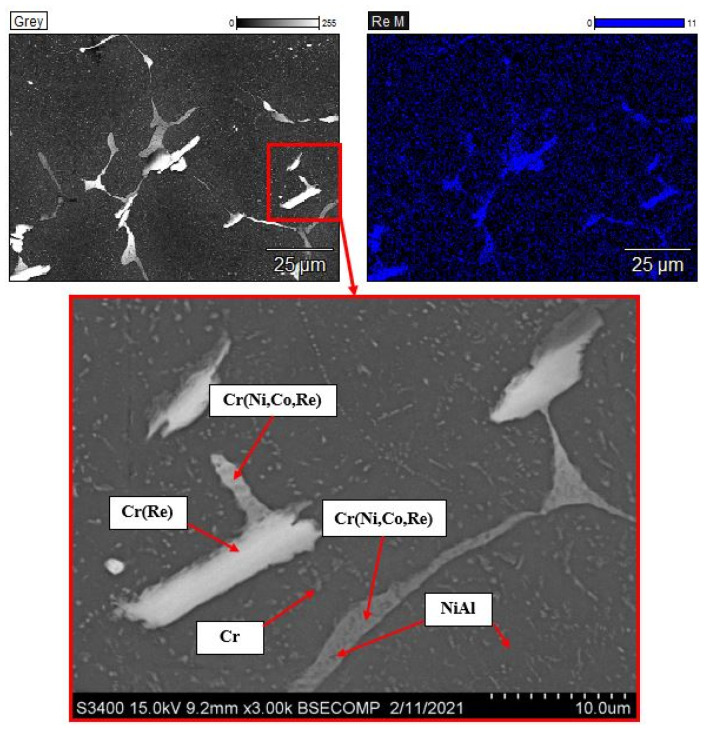
The microstructure of the alloy doped with rhenium (1.5%) and the map of dopant distribution.

**Figure 10 materials-14-03144-f010:**
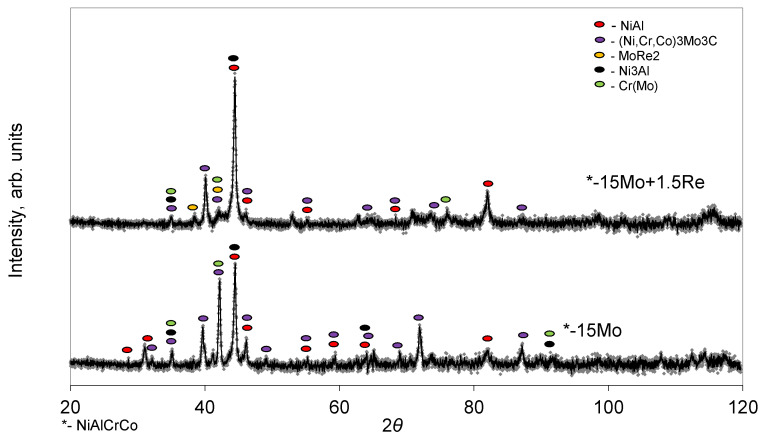
The XRD pattern of the NiAl–Cr–Co–(*X*) alloys, *X*= 15%Mo and 15%Mo + 1.5%Re.

**Figure 11 materials-14-03144-f011:**
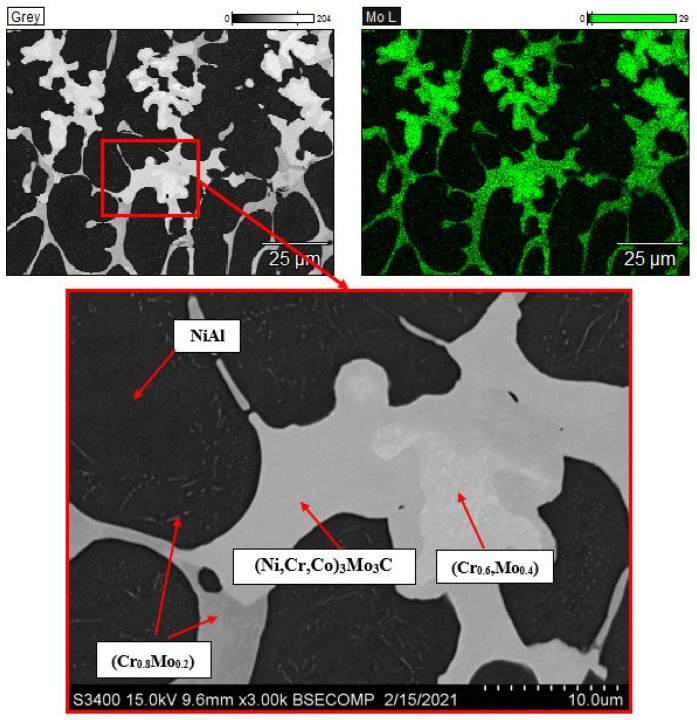
The microstructure of the alloy doped with 15% molybdenum and the map of doping element distribution.

**Figure 12 materials-14-03144-f012:**
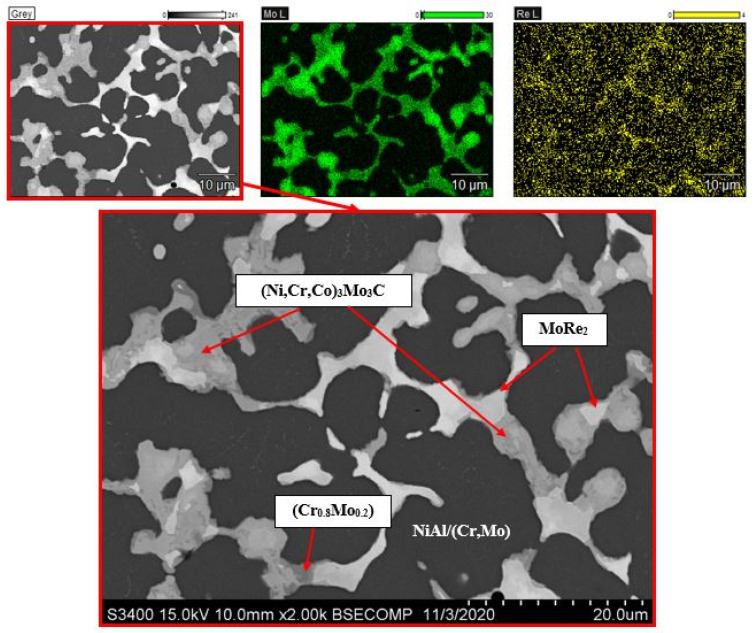
The microstructure of the alloy doped with 15% Mo and 1.5% Re and the map of dopant distribution.

**Figure 13 materials-14-03144-f013:**
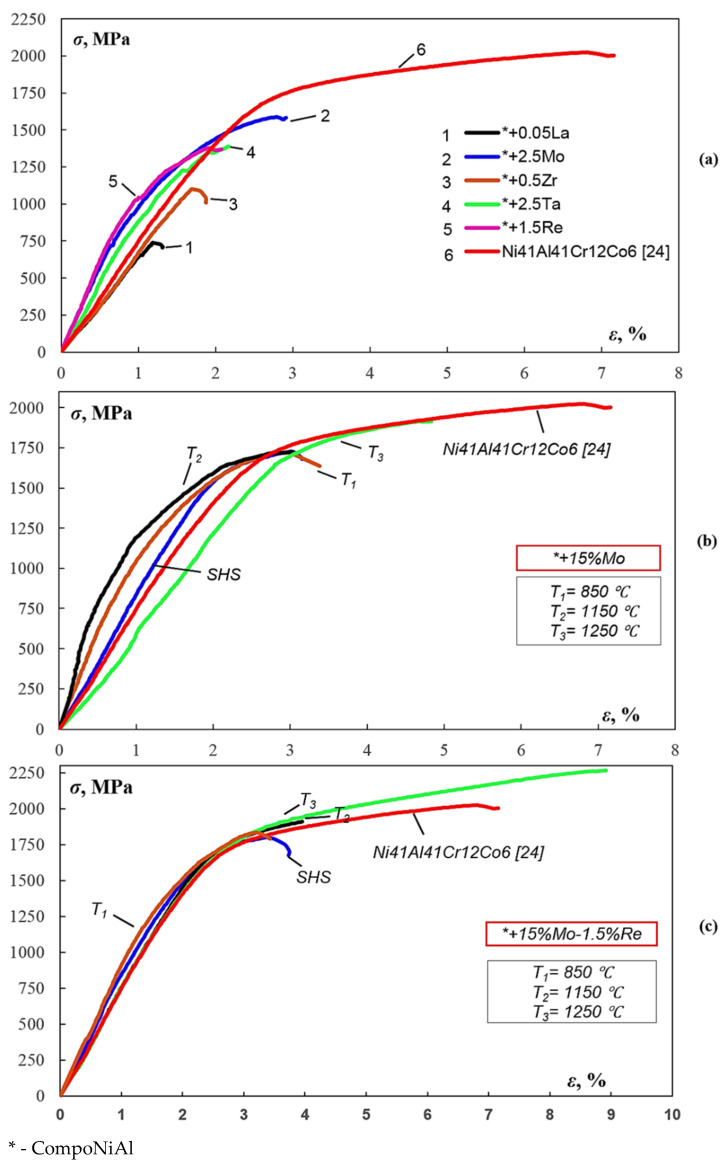
The compressive stress–strain curves of the CompoNiAl+ alloy (**a**) and the alloys doped with 15%Mo (**b**) and 15%Mo + 1.5%Re (**c**).

**Figure 14 materials-14-03144-f014:**
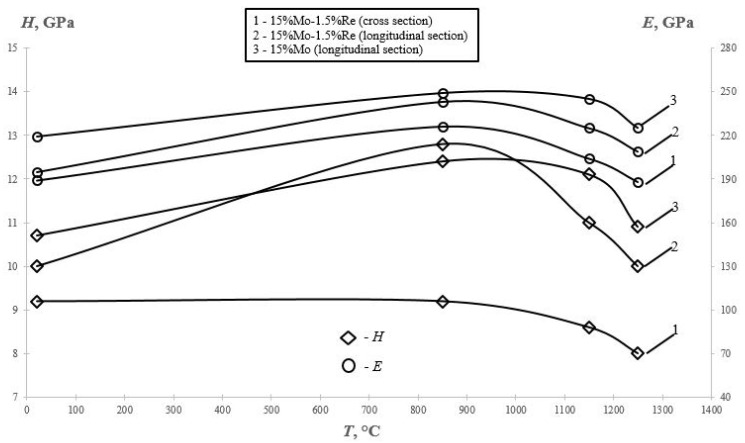
Mechanical properties of the NiAl–Cr–Co+15%Mo and NiAl–Cr–Co+15%Mo–1.5% Re alloy samples (in the longitudinal and transverse cross-sections) as a function of the annealing temperature according to the instrumented indentation data.

**Figure 15 materials-14-03144-f015:**
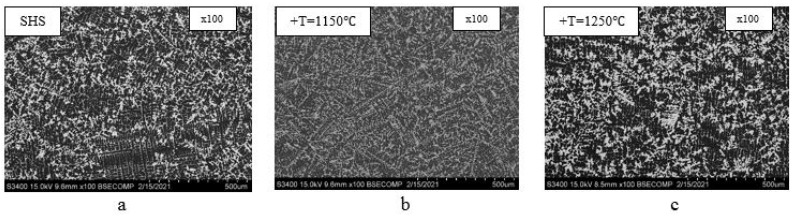
The microstructures of the NiAl–Cr–Co+15%Mo alloy before (**a**) and after annealing at 1150 °C (**b**) and 1250 °C (**c**).

**Figure 16 materials-14-03144-f016:**
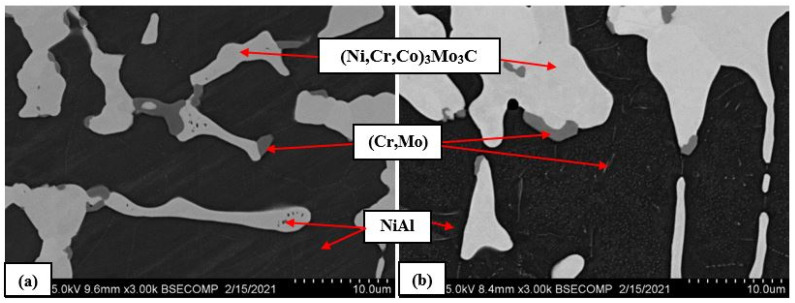
The microstructure of the NiAl–Cr–Co+15%Mo alloy after annealing at 1150 °C (**a**) and 1250 °C (**b**).

**Figure 17 materials-14-03144-f017:**
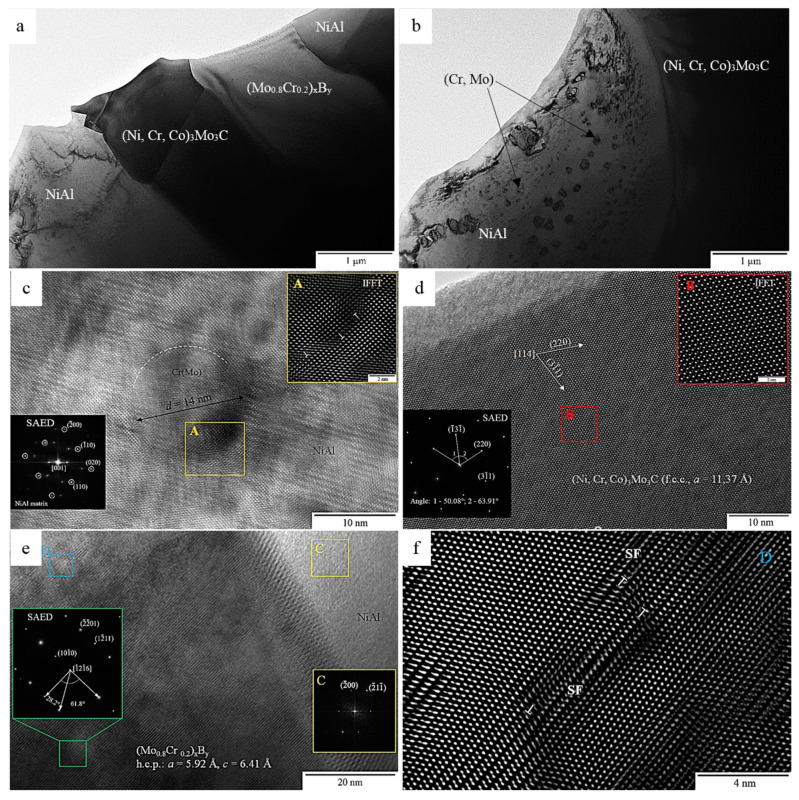
(**a**,**b**) The bright-field transmission electron microscopy (BF TEM) structure of the interface between the *β*-NiAl and Mo-containing phases. (**c**) The high-resolution TEM (HRTEM) image of the fine structure of *β*-NiAl dendritic cell with a Selected Area Electron Diffraction (SAED) inset taken along the [001] zone axis. The Inverse Fast Fourier Transformation (IFFT) filtered image of the area marked with A is shown in the upper right inset. (**d**) The HRTEM structure of the (Ni,Co,Cr)_3_Mo_3_C grain with a SAED inset taken along the [114] zone axis. The IFFT filtered area marked with B is shown in the upper right inset. (**e**) The fine structure of the NiAl/(Mo_0.8_Cr_0.2_)_x_B_y_ interface with a SAED inset taken along the [1216] zone axis. The Fast Fourier Transformation (FFT) filtered image of the area marked with C is shown in the lower right inset. (**f**) The IFFT filtered image of the fault crystal structure from the area is marked with D in Figure 17e.

**Figure 18 materials-14-03144-f018:**
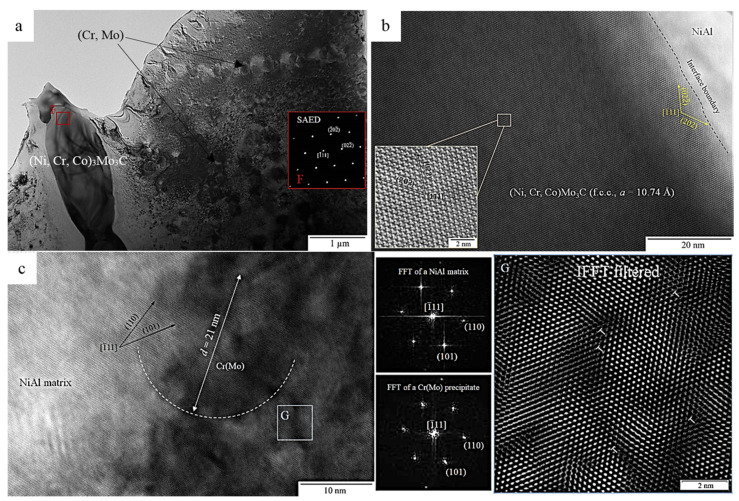
(**a**) The BF TEM structure of the NiAl-15Mo-12Cr-6Co alloy after annealing at 1250 °C with a [111] SAED inset (marked with F) from (Ni,Co,Cr)_3_Mo_3_C phase. (**b**) The HRTEM image of the crystal structure of the (Ni,Co,Cr)_3_Mo_3_C phase near the interface with the *β*-NiAl dendritic cell. (**c**) The crystal structure of *β*-NiAl cell and Cr(Mo) precipitate along the [111] zone axis. The FFT pattern from the *β*-NiAl matrix and Cr(Mo) precipitate. The [111] IFFT filtered area is marked with G.

**Figure 19 materials-14-03144-f019:**
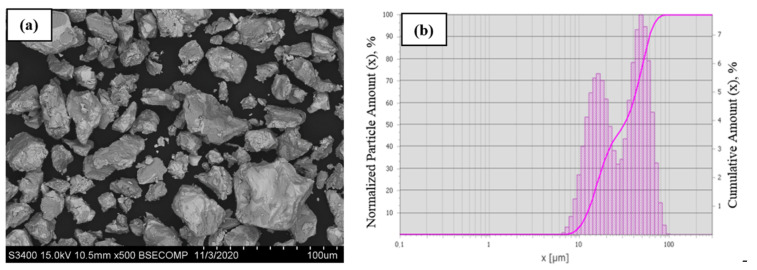
The morphology of the comminuted particles (**a**) and the granulometric composition (**b**) of the NiAl–Cr–Co+15%Mo alloy.

**Figure 20 materials-14-03144-f020:**
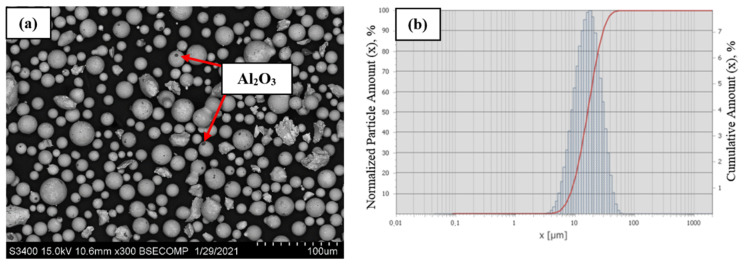
The morphology (**a**) and granulometric composition (**b**) of the NiAl–Cr–Co+15%Mo alloy powder at *I*_pl_ = 1.71 kW h/m^3^, *G*_powder_ = 3 kg/h, with Ar used as a plasma supporting gas.

**Figure 21 materials-14-03144-f021:**
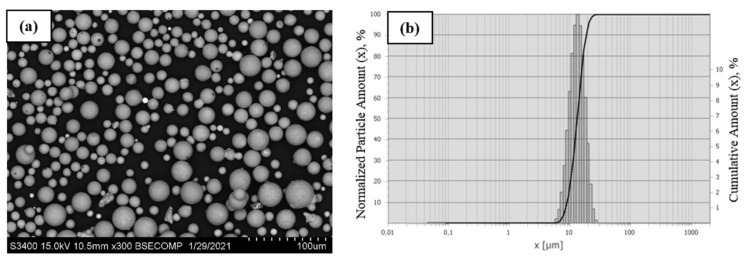
The morphology (**a**) and granulometric composition (**b**) of the NiAl-Cr-Co+15%Mo alloy powder at I_pl_ = 1.9 kW h/m^3^, G_powder_ = 3 kg/h, with Ar + H_2_ used as a plasma supporting gas.

**Figure 22 materials-14-03144-f022:**
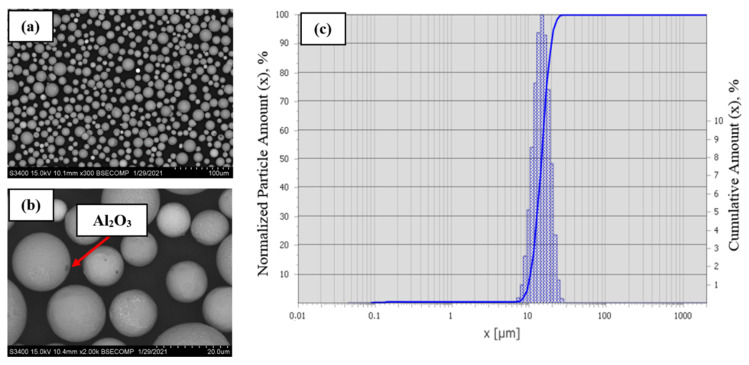
The morphology (**a**,**b**), differential, and integral distribution (**c**) of the spherical powder of the NiAl–Cr–Co+15%Mo alloy produced in the efficient treatment regime.

**Figure 23 materials-14-03144-f023:**
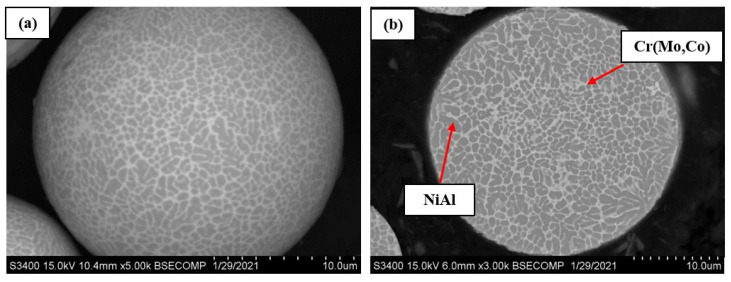
The microstructure of the surface (**a**) and the transverse cross-section (**b**) spherical powder of the NiAl–Cr–Co+15%Mo alloy.

**Table 1 materials-14-03144-t001:** The characteristics of the initial compounds and modifying agents.

Compound	Grade	Standard/ Specifications	Particle Size, µm	Chemical Composition, %
**Main components**				
NiO	special purity grade	TU 6-09-02439-87	<40	99.0
Cr_2_O_3_	pure	TU 6-09-4272-84	<20	99.0
Co_3_O_4_	special purity grade	GOST 18671-73	<30	99.0
Al	PA-4	GOST 6058-73	<140	98.0
Al	ASD-1	TU 48-5-226-87	<50	99.7
**Modifying agents (MA)**				
MoO_3_	pure for analysis	TU 6-09-4471-77	<50	99.0
Zr	E635	TU 95.166-83	⩽600	99.8
Ta	TaPM	TU 48-19-72-92	<20	98.0
Re	Re-0	TU 48-4-195-87	<150	99.99
La	LaM-1	TU 48-4-218-72	⩽600	99.85
B	Boron	TU 113-12-11.106-88	⩽90	98.6

**Table 2 materials-14-03144-t002:** The calculated composition of the alloy and the content of doping agents.

NiAl–Cr–Co *(basic alloy),* wt.%		Doping Agents (*X*), wt.%	
**Ni**	main element	**La**	0.3
		**Mo**	2.5–15.0
**Al**	22.5	**Zr**	0.5
**Co**	8.0	**Ta**	2.5
**Cr**	13.5	**Re**	1.5

**Table 3 materials-14-03144-t003:** The chemical composition of the NiAl–Cr–Co –(*X*) alloys.

Concentration, wt.%/at. %					*X* wt.%/at. %	
Ni	Al	Cr	Co	B		
54/43	22.30/37.90	14.01/12.35	8.34/6.49	0.015/0.06	**La**	0.083/0.03
	22.04/37.46	13.67/12.05	8.05/6.26	0.017/0.07	**Mo**	2.44/1.18
	22.68/38.55	13.03/11.49	8.33/6.48	0.015/0.06	**Zr**	0.48/0.24
	21.78/37.02	14.17/12.50	8.02/6.24	0.011/0.05	**Ta**	2.11/1.07
	21.91/37.24	13.98/12.33	8.18/6.36	0.016/0.07	**Re**	1.48/0.75

**Table 4 materials-14-03144-t004:** Impurity content in a self-propagating high-temperature synthesis (SHS) ingot of the NiAl–Cr–Co–(*X*) alloys.

*X*	Concentration, wt.%							
	W	C	Si	Fe	P	S	N	O
**La**	0.0273	0.011	0.053	0.094	0.0025	0.0128	0.00342	0.0210
**Mo**	0.0257	0.011	0.044	0.110	0.0029	0.0126	0.00368	0.0321
**Zr**	0.0257	0.013	0.079	0.102	0.0023	0.0126	0.00351	0.0216
**Ta**	0.0321	0.010	0.085	0.102	0.0024	0.0131	0.00296	0.0334
**Re**	0.0284	0.008	0.071	0.102	0.0024	0.0131	0.00305	0.0327

**Table 5 materials-14-03144-t005:** The phase composition and mechanical properties of the cast NiAl–Cr–Co–(*X*) alloys and individual phases.

Alloy	Phase	Phase Content, %	Lattice Parameter, Å	Sample		Phase	
				*H*, GPa	*E*, GPa	*H*, GPa	*E*, GPa
NiAl-Cr-Co+2.5%Mo	NiAl	100.0	2.879	8.2 ± 1.3	187 ± 20	8.2 ± 1.3	187 ± 20
NiAl-Cr-Co+0.5%Zr	NiAl	94.7	2.887	7.6 ± 1.3	173 ± 23	8.1 ± 0.5	181 ± 8
	ZrNi_5_	5.3	6.687			4.6 ± 0.2	119 ± 14
NiAl-Cr-Co+2.5%Ta	NiAl	85.9	2.894	10.6 ± 4.9	207 ± 40	9.2 ± 0.6	191 ± 7
	Ni_0.92_Ta_0.08_	9.2	3.584			10.6 ± 0.4	208 ± 9
	(Al_0.72_Ta_0.28_)Ni_3_	5.0	3.604			31.3	353
NiAl-Cr-Co+1.5%Re	NiAl	100.0	2.875	11.4 ± 6.0	199 ± 57	9.5 ± 0.2	181 ± 6
	Al(Re,Ni)_3_	-	-			30.1	378

**Table 6 materials-14-03144-t006:** Elemental composition of the NiAl–Cr–Co+15%Mo and NiAl–Cr–Co+15%Mo+1.5%Re alloys.

* *X*	Concentration, wt.%/at. %													
	Ni		Al		Cr		Co		Mo		B		Re	
15%Mo	47 ± 1.0	40 ± 1.0	19.6	35.94	11.3	10.75	6.11	5.13	15.2	7.84	0.015	0.07	-	
15%Mo-1.5%Re	45 ± 1.0	38 ± 1.0	18.7	34.88	12.3	11.90	6.8	5.81	15.40	8.08	0.019	0.09	1.49	0.4

*** NiAl-Cr-Co-(*X*).

**Table 7 materials-14-03144-t007:** The content of impurities in the NiAl–Cr–Co+15%Mo and NiAl–Cr–Co+15%Mo+1.5%Re alloys.

Matrix	Concentration, wt.%							
	W	C	Si	Fe	P	S	N	O
15%Mo	0.0211	0.011	0.045	0.092	0.0025	0.0124	0.00359	0.0215
15%Mo-1.5%Re	0.0218	0.011	0.041	0.114	0.0029	0.0121	0.00317	0.0188

Bi ≤ 0.0005; Sn ≤ 0.0001; Pb ≤ 0.0003 for all the systems under study.

**Table 8 materials-14-03144-t008:** The phase composition of the NiAl–Cr–Co–(*X*) alloys.

Alloy	Phase	Phase Content, %	Lattice Parameters, Å		
			a	b	c
NiAl-Cr-Co+15%Mo	NiAl	60.0	2.897	-	-
	(Ni,Cr,Co)_3_Mo_3_C	31.2	11.093	-	-
	Ni_3_Al	5.1	3.756	-	3.276
	(Cr, Mo)	3.7	3.118	-	-
NiAl-Cr-Co+15%Mo-1.5%Re	NiAl	54.5	2.866	-	-
	Ni_3_Al	6.6	3.765	-	3.270
	(Ni,Cr,Co)_3_Mo_3_C	32.4	11.081	-	-
	MoRe_2_	2.8	9.579	-	4.974
	(Cr, Mo)	3.7	3.035	-	-

**Table 9 materials-14-03144-t009:** Properties of the as-cast SHS alloys NiAl–Cr–Co–(*X*), where *X* = La, Mo, Zr, Ta, and Re.

No.	* +*X*	*T*_melt_, °C	*ρ*, g/cm^3^	*C*_v_, J/kg K	*σ*_ucs_, MPa	*σ*_ys_, MPa	*ε_pd_*, %
1	0.3La	1570	6.36	636	738	-	˂1 **
2	2.5Mo	1580	6.44	644	1586	-	˂1 **
3	0.5Zr	1600	6.37	680	1099	-	˂1 **
4	2.5Ta	1590	6.49	671	1388	-	˂1 **
5	1.5Re	1585	6.55	580	1378	-	˂1 **
6	15Mo	1580	7.06	706	1728	1566	0.95
7	15Mo-1.5Re	1585	7.25	615	1800	1618	1.10

* The NiAlCr(Co)+ matrix; ** Brittle failure of the samples occurred.

**Table 10 materials-14-03144-t010:** Mechanical properties during the compression tests at *T* = 20 °C.

No.	* +*X*	*σ*_ucs_, MPa	*σ*_ys_, MPa	*ε*_pd_, %
**1.1**	**15Mo**	1728	1566	0.95
1.2	*T =* 850 °C*, t* = 180 min	1721	1636	˂1 **
1.3	*T =* 1150 °C*, t* = 180 min	1726	1642	˂1 **
1.4	*T =* 1250 °C*, t* = 180 min	1916	1653	2.01
**2.1**	**15Mo-1.5Re**	1800	1618	1.10
2.2	*T =* 850 °C, *t* = 180 min	1833	1628	1.98
2.3	*T =* 1150 °C, *t =* 180 min	1910	1634	1.59
2.4	*T =* 1250 °C, *t =* 180 min	2267	1740	6.15

* The NiAlCr(Co)+ matrix; ** Brittle failure of the samples occurred.

**Table 11 materials-14-03144-t011:** The composition of the structural components of the alloy under study, at. %.

EDXS Area					
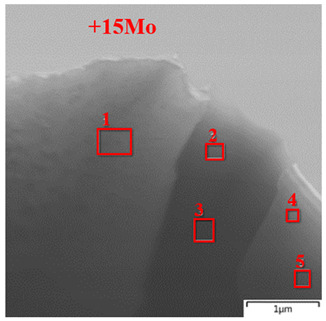			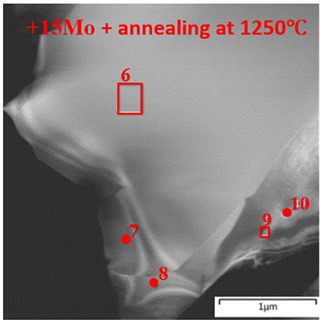		
Spectrum	Ni	Al	Cr	Co	Mo
+15Mo					
1	50.77	35.75	6.37	6.38	0.74
2	18.87	-	22.86	7.48	50.78
3	20.66	-	22.41	7.60	49.33
4	-	-	23.31	-	76.69
5	-	-	20.56	-	79.44
+15Mo + annealing at 1250 °C					
6	21.13	-	24.81	9.62	44.44
7	23.95	-	22.02	9.18	44.85
8	20.53	-	35.70	13.72	30.05
9	-	-	84.66	-	15.34
10	50.38	39.71	4.67	5.24	-

## Data Availability

Data is contained within the article.

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
