# Peer review of "Structure and Properties of Heat-Resistant Alloys NiAl–Cr–Co–X (X = La, Mo, Zr, Ta, Re) and Fabrication of Powders for Additive Manufacturing"

_materials, 2021, doi:10.3390/ma14123144_

Round 1
Reviewer 1 Report
Dear author, I have carefully studied your article and must make my comments:
- The volume of the article is very large and reminds me of a laboratory study report in the former USSR.
- It is advisable to separate the results and the discussion for a better understanding by readers of the features of the process, enter additionally the names of the research segments and descriptions of the processes.
- For the use of the materials in aviation and astronautics, it is also necessary to study tensile stresses with temperature changes. You only measured the mechanical properties under compression, which is not enough. The hardness of the various phases had to be measured by using the nano-indentation by the Berkovich indenter, this gives more accurate results in comparison with the calculation.
- Why didn't you use the addition of carbon to the initial powder in the SHS process? You have shown that the grain size affects the temperature of the onset and the rate of the SHS process, as well as the formation of the structure and the appearance of new phases. It is known that carbon carbides mainly affect compressive strength.
For a better understanding of your work by the readers, I advise you to make these insignificant changes in the text of the study.
Reviewer 2 Report
Dear Authors,
in my opinion, your manuscript entitled: "Structure and properties of heat-resistant alloys NiAl-Cr-Co-X (X = La, Mo, Zr, Ta, Re) and fabrication of powders for additive manufacturing" can be published in Materials after minor revision.
Please, can you add in the Introduction more details about the novelty of your work and the main goal of your study?
Could you show the image of the microstructure of the alloy doped with rhenium and its profile of element distribution in the intergrain space (as like in Figure 8 for the alloy doped with tantalum)?
You have to improve the STEM images in Table 11, because in the present form is difficult to estimate for which areas, the selected EDX spectra are presented.
Kind regards,
Reviewer
